# Association of *N-acetyltransferases 1 and 2* Polymorphisms with Susceptibility to Head and Neck Cancers—A Meta-Analysis, Meta-Regression, and Trial Sequential Analysis

**DOI:** 10.3390/medicina57101095

**Published:** 2021-10-13

**Authors:** Hady Mohammadi, Mehrnoush Momeni Roochi, Masoud Sadeghi, Ata Garajei, Hosein Heidar, Bayazid Ghaderi, Jyothi Tadakamadla, Ali Aghaie Meybodi, Mohsen Dallband, Sarton Mostafavi, Melina Mostafavi, Mojtaba Salehi, Dena Sadeghi-Bahmani, Serge Brand

**Affiliations:** 1Department of Oral and Maxillofacial Surgery, Fellowship in Maxillofacial Trauma, Health Services, Kurdistan University of Medical Sciences, Sanandaj 6617713446, Iran; hadi.mohammadi@muk.ac.ir; 2Department of Oral and Maxillofacial Surgery, Fellowship in Maxillofacial Trauma, School of Dentistry, Tehran University of Medical Sciences, Tehran 1439955991, Iran; mehrnoushmomeni@yahoo.com (M.M.R.); atagarajei@tums.ac.ir (A.G.); drheidar@yahoo.com (H.H.); mojtabasalehi9241@gmail.com (M.S.); 3Department of Biology, Science and Research Branch, Islamic Azad University, Tehran 1477893855, Iran; sadeghi_mbrc@yahoo.com; 4Department of Head and Neck Surgical Oncology and Reconstructive Surgery, The Cancer Institute, School of Medicine, Tehran University of Medical Sciences, Tehran 1439955991, Iran; ali.aghaimeybodi@gmail.com; 5Cancer and Immunology Research Center, Department of Internal Medicine, School of Medicine, Kurdistan University of Medical Sciences, Sanandaj 6617913446, Iran; Bayazidg@yahoo.com; 6School of Medicine and Dentistry, Griffith University, Brisbane, QLD 4222, Australia; j.tadakamadla@griffith.edu.au; 7Department of Oral and Maxillofacial Surgery, Dental School, Taleghani Hospital, Shahid Beheshti University of Medical Sciences, Tehran 1983963113, Iran; m-dalband@sbmu.ac.ir; 8English Department, Baneh Branch, Islamic Azad University, Baneh 6691133845, Iran; sartonmostafave@yahoo.com; 9Tehran Medical Branch, Islamic Azad University, Tehran 1419733171, Iran; mostafavi.melina@yahoo.com; 10Sleep Disorders Research Center, Kermanshah University of Medical Sciences, Kermanshah 6719851115, Iran; bahmanid@stanford.edu; 11Center for Affective, Stress and Sleep Disorders, University of Basel, Psychiatric Clinics, 4001 Basel, Switzerland; 12Substance Abuse Prevention Research Center, Kermanshah University of Medical Sciences, Kermanshah 6715847141, Iran; 13Department of Psychology, Stanford University, Stanford, CA 94305, USA; 14Division of Sport Science and Psychosocial Health, Department of Sport, Exercise and Health, University of Basel, 4052 Basel, Switzerland; 15School of Medicine, Tehran University of Medical Sciences, Tehran 1416753955, Iran

**Keywords:** head and neck carcinoma, oral carcinoma, polymorphism, N-acetyltransferases, meta-analysis

## Abstract

*Background and objective:**N-acetyltransferases 1 and 2* (*NAT1* and *NAT2*) genes have polymorphisms in accordance with slow and rapid acetylator phenotypes with a role in the development of head and neck cancers (HNCs). Herein, we aimed to evaluate the association of *NAT1* and *NAT2* polymorphisms with susceptibility to HNCs in an updated meta-analysis. *Materials and methods:* A search was comprehensively performed in four databases (Web of Science, Scopus, PubMed/Medline, and Cochrane Library until 8 July 2021). The effect sizes, odds ratio (OR) along with 95% confidence interval (CI) were computed. Trial sequential analysis (TSA), publication bias and sensitivity analysis were conducted. *Results*: Twenty-eight articles including eight studies reporting *NAT1* polymorphism and twenty-five studies reporting *NAT2* polymorphism were involved in the meta-analysis. The results showed that individuals with slow acetylators of *NAT2* polymorphism are at higher risk for HNC OR: 1.22 (95% CI: 1.02, 1.46; *p* = 0.03). On subgroup analysis, ethnicity, control source, and genotyping methods were found to be significant factors in the association of *NAT2* polymorphism with the HNC risk. TSA identified that the amount of information was not large enough and that more studies are needed to establish associations. *Conclusions:* Slow acetylators in *NAT2* polymorphism were related to a high risk of HNC. However, there was no relationship between *NAT1* polymorphism and the risk of HNC.

## 1. Introduction

Cellular inflammation and immunity can play a significant role in various stages of carcinogenesis [1] such as head and neck cancers (HNCs). HNC mortality rates are elevating and disproportionately affect people in low- and middle-income countries and areas with restricted resources [2]. Global Burden of Disease Study (GBD) in 2016 estimated 512,492 deaths due to HNC (a minimum of 15,018 deaths in North Africa and the Middle East to a maximum of 199,280 in South Asia) and predicted the death count to reach 705,901 in 2030 [3,4]. HNC involves a series of tumors originating in the oropharynx, hypopharynx, oral cavity, lip, larynx, or nasopharynx [5]. Smoking, alcohol consumption, and high-risk human papillomaviruses have been related to HNC [5,6,7]. In connection with the role of genetics in HNC, several recent meta-analyses have reported the association of polymorphisms with the risk of HNCs [8,9,10,11].

A number of heterocyclic and aromatic amines are the main carcinogenic compounds of tobacco smoke [12,13] that their metabolism in humans is complex and includes acetylation as a main pathway for DNA mutation and the onset of carcinogenesis [14]. In particular, two N-acetyltransferases, *NAT1* and *NAT2* perform a role in catalyzing the deactivation and activation of several carcinogenic amines through N- and O-acetylation, respectively [14,15]. Both *NAT* genes (*NAT1* and *NAT2*) have polymorphisms in humans and in accordance with slow and rapid acetylator phenotypes [16]. The *NAT2* metabolized gene is located in region 10 of chromosome 8p21, which contains two exons with a long intron of about 8.6 kb [17]. Exon 1 is very short (100 bp) and the entire protein-coding region in Exon 2 is 870 bp [18]. Also, the *NAT1* gene is located on the short arm of chromosome 8 (8p21) [19,20]. *NAT1* accelerates acetylation specifically for arylamine receptor structures such as p-aminosalicylic and p-aminobenzoic acids [21] and *NAT2* acetylates other arylamine-acceptor structures, such as isoniazid, sulfasalazine, procainamide, and caffeine [19].

Evidence from the published articles on the relationship between *NAT1* and *NAT2* polymorphisms and HNC susceptibility is conflicting [22,23]. The association between the polymorphisms (*NAT1* and *NAT2*) and the HNC risk has been evaluated by one [24] and four [25,26,27,28] meta-analyses, respectively. However, these studies were published several years ago with the most recent one being published in 2015. Therefore, through this meta-analysis, we intend to update the evidence on the association between the polymorphisms and the HNC risk by including more studies. In addition, we aim to conduct trial sequential analysis (TSA) and meta-regression.

## 2. Materials and Methods

### 2.1. Study Design

The present meta-analysis follows the Preferred Reporting Items for Systematic Reviews and Meta-Analyses (PRISMA) protocols [29]. The PI/ECO (population, intervention/exposure, comparison, and outcome) question was: Are polymorphisms of *NAT1* and *NAT2* associated with the risk of HNC?

### 2.2. Identification of Articles

A search was comprehensively performed by one author (M.S.) in four databases of Web of Science, Scopus, PubMed/Medline, and Cochrane Library until 8 July 2021, without any restrictions in language, publication year, age, and sex to retrieve the relevant articles (Figure 1). The titles and abstracts of the relevant articles were assessed by the same author (M.S.); subsequently, the full-texts of the articles found to be relevant based on the eligibility criteria were downloaded. The search strategy included: (“N-acetyl transferases” or “N-acetyltransferase” or “*NAT2*” or “*NAT1*”) and (“mouth” or “OSCC” or “oral” or “tongue” or “head and neck” or “HNSCC“ or “nasopharyngeal” or “nasopharynx” or “oropharyngeal” or “salivary gland” or “laryngeal” or “larynx” or “hypopharyngeal” or “pharyngeal” or “pharynx” or “oral cavity” or “hypopharynx”) and (“tumor” or “carcinoma” or “cancer” or “neoplasm”) and (“allele” or “variant” or “polymorphism” or “genotype” or “gene”). The reference lists of the retrieved articles were reviewed to ensure that no important study was missed. Another author (H.M.) re-checked the process of searching and article selection. A lack of agreement between both authors was resolved by another author (J.T.).

### 2.3. Eligibility Criteria

The inclusion criteria were: (1) case-control studies reporting slow and rapid acetylators of *NAT1* and *NAT2* polymorphisms in HNC patients and controls, (2) HNC patients were diagnosed clinically and pathologically, and (3) HNC patients had no other systemic diseases and controls were healthy or free of tumors. On the contrary, meta-analyses, review studies, articles with incomplete data, studies without a control group, animal studies, conference papers, book chapters, and comment papers were excluded.

### 2.4. Data Summary

The data of the articles involved in the meta-analysis were separately retrieved by two authors (M.S. and S.B.). Extracted data included names of the authors, publication year, study country, ethnicity, number of cases, tumor type, source of controls, genotyping method, quality score, age, and gender distribution.

### 2.5. Quality Evaluation

The quality scoring was completed by one author (M.S.) based on the Newcastle-Ottawa Scale (NOS) scale [30] that a study is judged on three broad perspectives: the selection (4 scores); the comparability (2 scores); and the outcome (3 scores) for non-randomized studies, respectively. The maximum possible score was nine and high-quality studies were those with a score of ≥7.

### 2.6. Statistical Analysis

The effect sizes, odds ratios (OR) along with 95% confidence interval (CI), were calculated using the Review Manager 5.3 (RevMan 5.3; the Cochrane Collaboration, the Nordic Cochrane Centre, Copenhagen, Denmark) as well as subgroup analyses, quantifying the association between *NAT1* and *NAT2* polymorphisms and the HNC risk. A *p*-value (2-sided) < 0.05 was considered as a significant value. A random-effects model [31] was performed when I^2^ statistic represented a significant heterogeneity (P_heterogeneity_ < 0.1 or I^2^ > 50%) and if the heterogeneity was insignificant, a fixed-effect model [32] was applied.

Subgroup analyses were performed based on the ethnicity of study participants, control source in the study, tumor type, sample size, and genotyping method used in a study. To adjust for the effect of sample sizes, gender, and age distribution of the subjects included in the studies, a meta-regression analysis was conducted.

Publication bias was assessed applying funnel plots, Egger’s and/or Begg’s tests with a *p*-value (2-sided) < 0.05 demonstrating the existence of publication bias. Sensitivity analyses (“one-study-removed” and “cumulative” analyses) were conducted to evaluate the stability of pooled ORs. The meta-regression, publication bias, and sensitivity analysis were analyzed using the Comprehensive Meta-Analysis version 2.0 (CMA 2.0) software (CMA 2.0; Biostat Inc., Englewood, NJ, USA).

To illustrate false-positive or negative conclusions from meta-analyses [33], trial Sequential Analysis (TSA) software (version 0.9.5.10 beta) (Copenhagen Trial Unit, Centre for Clinical Intervention Research, Rigshospitalet, Copenhagen, Denmark) was used to evaluate TSA for analyses [34]. A futility threshold can be checked by the TSA to determine the effectiveness or ineffectiveness before information size is reached. The required information size (RIS) and a two-sided boundary type were computed with an alpha risk of 5% and beta risk of 20%. There were enough studies where the Z-curve reached the RIS line or the boundary line or entered the futility area. Otherwise, the amount of information was not enough and more evidence was needed.

### 2.7. Primer Sequences

The primer sequences of *NAT1* and *NAT2* are shown in the studies of Katoh et al. [35] and Chen et al. [36], respectively.

## 3. Results

### 3.1. Study Selection

From the four electronic databases and manual searching, 265 records were identified. After excluding the duplicates and irrelevant records, 48 full-text articles met the eligibility criteria (Figure 1). Then, 20 full-texts were removed (five were meta-analyses, one was an umbrella review, three were reviews, one was a book, one was an animal study, five articles had no control groups, one article had insufficient data, and three articles did not report genotypes of slow and rapid). Finally, 28 articles were used in the meta-analysis.

### 3.2. Characteristics of Studies

Twenty-eight studies included in the analysis were published between 1998 and 2014 (Table 1). Fourteen articles [22,23,36,37,38,39,40,41,42,43,44,45,46,47] reported the results in Caucasians, nine [35,48,49,50,51,52,53,54,55] in Asians, and five [56,57,58,59,60] among participants of mixed ethnicity. The control source in eighteen articles [22,23,35,37,39,40,43,44,45,46,48,49,51,52,53,57,58,60] was hospitals and ten [36,38,41,42,47,50,54,55,56,59] recruited the controls from a general population. In total, the articles included 5154 HNC cases and 6194 controls. Age, gender distribution, sample size, tumor type, genotyping method, and the quality score are shown in Table 1.

Table 2 shows the prevalence of slow and rapid acetylators of *NAT1* and *NAT2* polymorphisms. Eight studies [23,35,38,39,44,47,52,60] included *NAT1* polymorphism with 1509 HNC cases and 1829 controls and twenty-five studies [22,23,35,36,37,38,40,41,42,43,44,45,46,47,48,49,50,51,53,54,55,56,57,58,59] included *NAT2* polymorphism with 4393 HNC cases and 5321 controls.

### 3.3. Pooled Analyses

The pooled OR for the association between *NAT1* polymorphism and the risk of HNC from eight studies was 0.89 (95% CI: 0.77, 1.02; *p* = 0.09; I^2^ = 48%), (Figure 2). The pooled effect estimate was not significant demonstrating no association between *NAT1* polymorphism and the risk of HNC.

Forest plot in Figure 3 illustrates that the pooled OR was 1.22 (95% CI: 1.02, 1.46; *p* = 0.03; I^2^ = 74%) for the relationship between *NAT2* polymorphism and the HNC risk. This indicates that slow acetylators are related to high risk of HNC.

### 3.4. Subgroup Analyses

When there was one study for a subgroup, we could delete it [61]. Subgroup analyses were performed based on ethnicity, sample size, control source, genotyping method, and tumor type (Table 3). With regards to *NAT1* polymorphism, no subgroup differences were observed. For *NAT2* polymorphism, significant subgroup effects were observed for ethnicity and the control source. Slow acetylators among Asians and also the population-based studies could be effective factors on the pooled result of the association between *NAT2* polymorphism and the HNC risk.

### 3.5. Meta-Regression

The meta-regression analyses assessing the effect of publication year, the sample size, and the mean age and gender distribution of cases and controls on the risk of HNC in *NAT1* and *NAT2* polymorphisms are shown in Table 4. Sample size, the mean age of cases, and the percentage of males in the controls were confounding factors for the pooled result of the association between *NAT2* polymorphism and the HNC susceptibility. With an increase in sample size, age of the cases, and percentage of males in the controls, the OR decreased.

### 3.6. Trial Sequential Analysis

TSA for both polymorphisms (*NAT1* and *NAT2*) and the HNC risk is illustrated in Figure 4. The Z-curve (blue line) did not reach the RIS or the boundary lines or enter the futility area for either polymorphism and therefore, the amount of information was not large enough, suggesting the need for more studies.

### 3.7. Sensitivity Analysis

Both “one-study-removed” and “cumulative analysis” illustrated the pooled data stability for *NAT1* and *NAT2* polymorphisms (data not presented). After removing one study [46] with outlier data, in concordance with previous analysis, the new pooled result did not report any relationship between *NAT2* polymorphism and the HNC susceptibility (OR = 1.17; 95% CI: 0.99, 1.39; *p* = 0.07, I^2^ = 70%). In addition, after removing the studies with a quality score of less than 7 for *NAT2* polymorphism [43,49,53], the new result remained similar (OR = 1.27; 95% CI: 1.03, 1.57; *p* = 0.03; I^2^ = 76%). Removal of studies with a quality score of less than 7 for NAT1 polymorphism [39,52], did not change the pooled estimate (OR = 0.85; 95% CI: 0.75, 1.02; *p* = 0.08; I^2^ = 62%).

### 3.8. Publication Bias

The Egger’s (*p* = 0.240) and Begg’s (*p* = 0.322) tests did not reveal any publication bias for *NAT1* polymorphism, but both tests revealed the presence of publication bias for *NAT2* polymorphism (Egger’s: *p* = 0.012 and Begg’s: *p* = 0.028), (Figure 5).

## 4. Discussion

This meta-analysis showed a significant relationship between *NAT2* polymorphisms and the HNC susceptibility with slow acetylators being at higher risk for HNC than rapid acetylators. For *NAT2* polymorphism, the ethnicity, the control source, and genotyping methods could modify the association of this polymorphism and the HNC risk. In addition, TSA showed the amount of information for the association between the polymorphisms (*NAT1* and *NAT2)* and the HNC risk was not large enough.

The findings from studies exploring the association of *NAT1* polymorphism with other cancers and HNC are different. One meta-analysis [24] found *NAT1* polymorphism to be related to the risk of lung, colorectal, head and neck, bladder, and gastric carcinomas, but not with prostate, breast, and pancreatic carcinomas and non-Hodgkin’s lymphoma. Varzim et al. [47] checked the association between *NAT1* polymorphism and the laryngeal cancer risk and found that the association depends on tumor location. Among the eight studies included in our meta-analyses [23,35,38,39,44,47,52,60] which evaluated the association between *NAT1* polymorphism and the HNC risk, just one study [35] reported a protective role of *NAT1* slow acetylators in the HNC patients while the rest of the studies did not find any association.

Comparing the individual studies included in the meta-analysis, differences were observed between the studies. For example, five studies [41,46,48,55] found an elevated risk of HNC for *NAT2* slow acetylators, one found a protective role of these acetylators in HNC patients, and three did not find any association between *NAT2* polymorphism and the HNC risk [23,45,49].

Effective factors on the association between NAT polymorphisms and the risk of HNC were not included in our analysis due to low numbers of studies, including smoking, gene combination, and the linkage disequilibrium. One study [41] found an elevated frequency of the NAT2 slow acetylator genotypes among HNC patients who smoked less than those who smoked more frequently. Another study reported an association in cases with a smoking history ≤30 years in duration [35]. These contradictory results [35,41,46] suggest the need to evaluate the effect of NAT polymorphisms independent of the history of smoking. In addition, assessing the frequencies of gene-gene combination (*NAT2* with GSTM1, XPD, and CYP1A1) between cases with laryngeal cancer and the controls, the frequency of combinations was superior to cases than in controls where the numbers of combinations had an increased risk of laryngeal cancer and the numbers of other combinations had a protective role [40]. The linkage disequilibrium between the genes of *NAT1* and *NAT2* has been observed in HNC [23,38,62] and other cancers [63,64,65]. Research [66] showed the highest level of carcinogen-DNA adducts formation in cases with acetylation activity of *NAT1* rapid and *NAT2* slow. Therefore, future studies should consider the linkage between these polymorphisms.

The limitations of the present meta-analysis were: (1) low sample size in some studies. (2) In a number of the involved studies, the controls were not well matched to the cases. (3) Low numbers of studies entered to the analysis as shown by TSA. (4) Existence of publication bias and high heterogeneity between the analyses.

## 5. Conclusions

There was no association between *NAT1* polymorphism and susceptibility to HNC, whereas an association between and *NAT2* polymorphism and the HNC risk was found. Slow acetylators of *NAT2* polymorphism were at greater risk for HNC than the rapid acetylators. Despite the stability of the results, the presence of high heterogeneity, publication bias, and confounding factors warrant the need for more studies to confirm the results of the present meta-analysis as well as TSA.

## Figures and Tables

**Figure 1 medicina-57-01095-f001:**
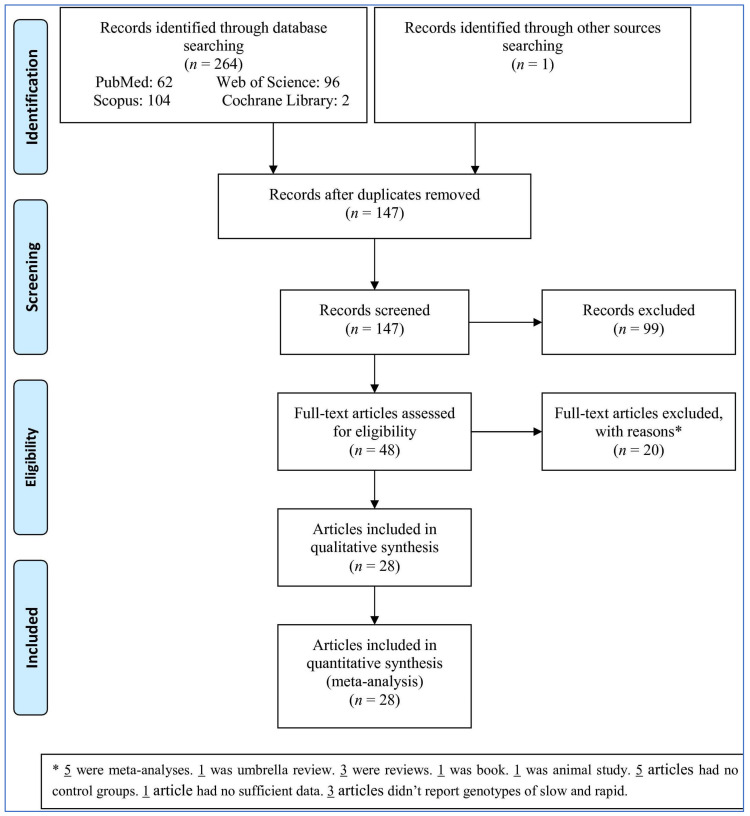
Flowchart of the study selection.

**Figure 2 medicina-57-01095-f002:**
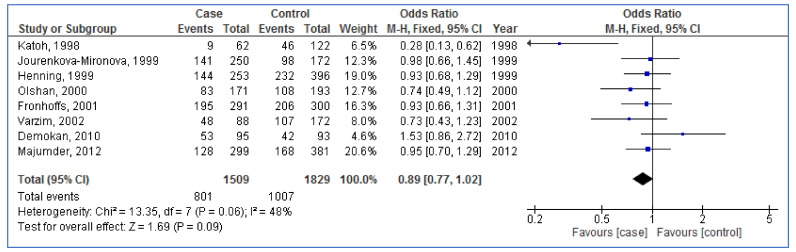
Forest plot for the association between *N-acetyltransferases 1* (*NAT1*) polymorphism and the risk of head and neck cancer (slow vs. rapid acetylators). The diamond at the bottom of the forest plot illustrates the pooled result. The square in front of a individual study shows the result of the study and its horizontal line shows 95% confidence interval of the result.

**Figure 3 medicina-57-01095-f003:**
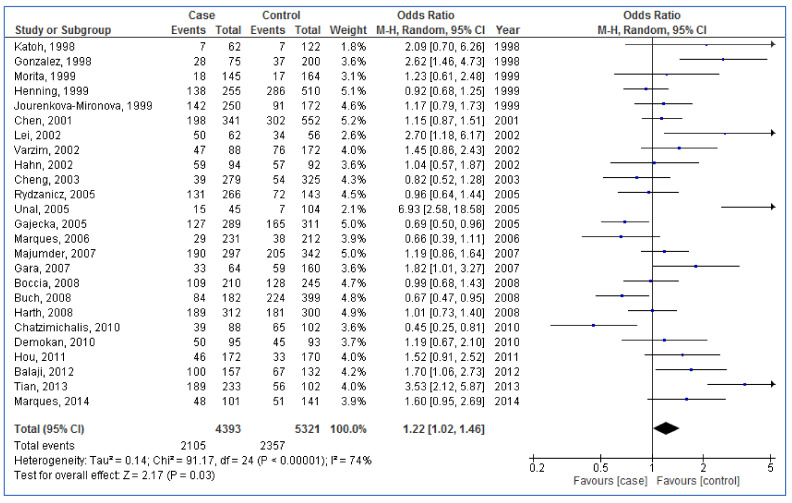
Forest plot demonstrating association between *N-acetyltransferases 2* (*NAT2*) polymorphism and the risk of head and neck cancer (slow vs. rapid). The diamond at the bottom of the forest plot illustrates the pooled result. The square in front of a individual study shows the result of the study and its horizontal line shows 95% confidence interval of the result.

**Figure 4 medicina-57-01095-f004:**
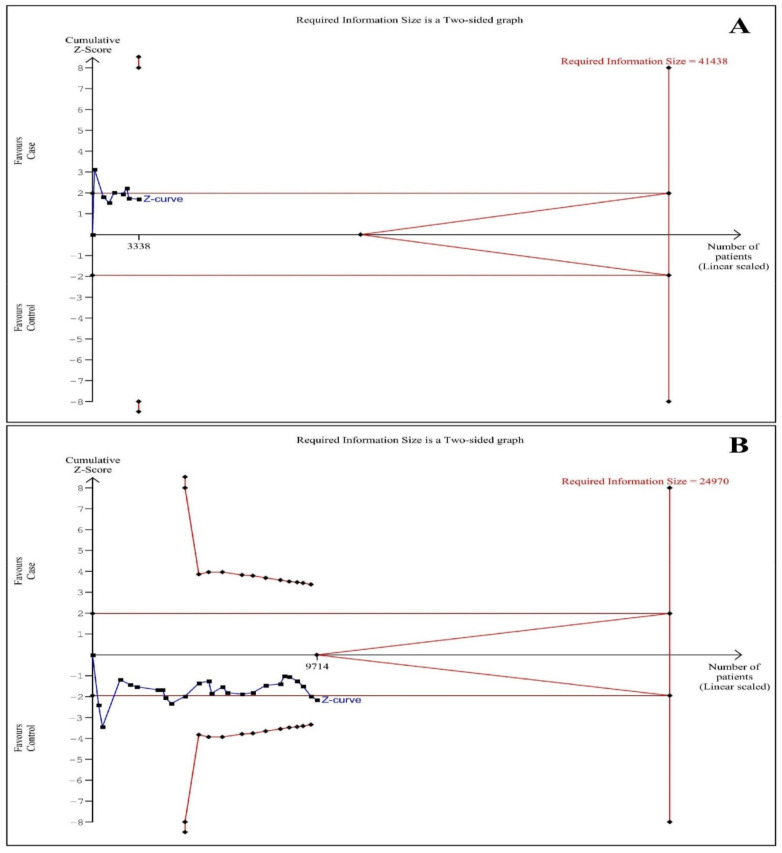
Trial sequential analysis of association between *N-acetyltransferases 1 and 2* (*NAT1* and *NAT2*) polymorphisms and the risk of head and neck cancer (slow vs. rapid acetylators) [α = 5% and 1-β = 80%]. (**A**) *NAT1* [diversity or D^2^ = 52%] and (**B**) *NAT2* [D^2^ = 76%].

**Figure 5 medicina-57-01095-f005:**
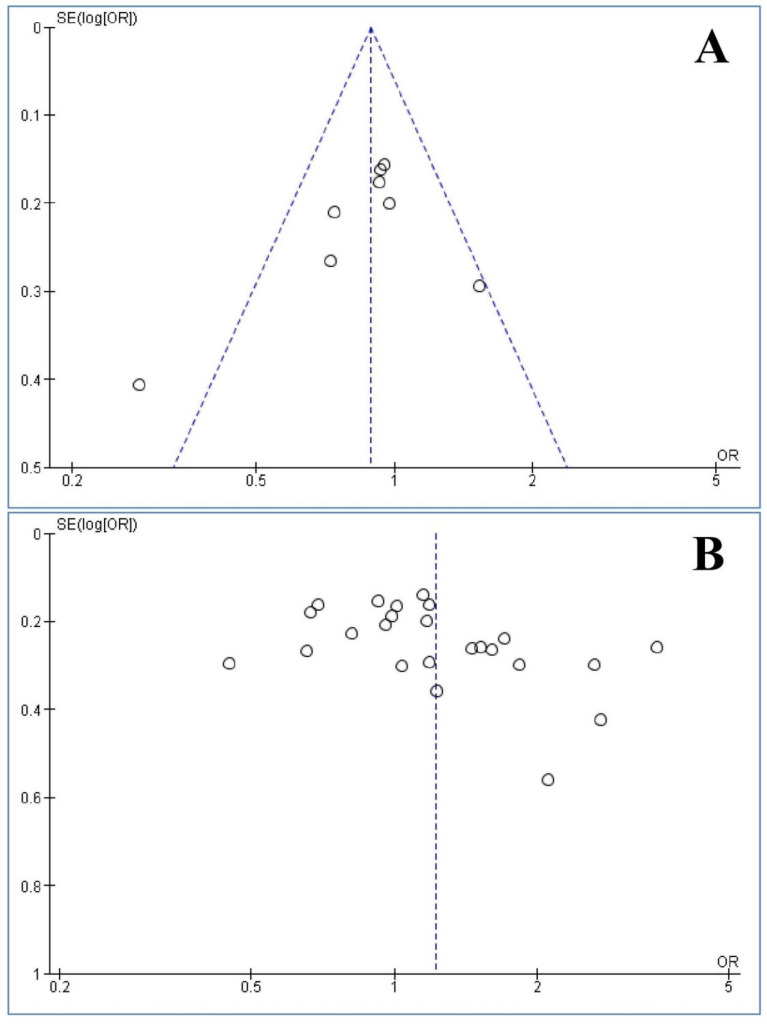
Funnel plot analyses of the association between *N-acetyltransferases 1 and 2* (*NAT1* and *NAT2*) polymorphisms and the risk of head and neck cancer (slow vs. rapid acetylators). (**A**) *NAT1* and (**B**) *NAT2*.

**Table 1 medicina-57-01095-t001:** Characteristics of the articles included in the meta-analysis.

The First Author, Publication Year	Country	Ethnicity	Control Source	Number		Mean Year		Male Percentage		Type of Tumor	Genotyping Method	Quality Score
				Case	Control	Case	Control	Case	Control			
Gonzalez, 1998 [41]	Spain	Caucasian	PB	75	200	58.7	45	100	75	Oral, pharyngeal, laryngeal	PCR-RFLP	7
Katoh, 1998 [35]	Japan	Asian	HB	62	122	61.7	62.4	64.5	61.5	Oral	PCR-RFLP	7
Henning, 1999 [23]	Germany	Caucasian	HB	255	510	61.4	NA	90.6	NA	Laryngeal	PCR	7
Jourenkova-Mironova, 1999 [44]	France	Caucasian	HB	250	172	54.4	54.9	96	94.8	Oral, pharyngeal, laryngeal	PCR-RFLP	7
Morita, 1999 [54]	Japan	Asian	PB	145	164	59.0	49.8	86.9	62.2	Oral, pharyngeal, laryngeal	PCR	7
Olshan, 2000 [60]	USA	Mixed	HB	171	193	59.5	56.8	81.3	59.1	Oral, pharyngeal, laryngeal	PCR	7
Chen, 2001 [36]	USA	Caucasian	PB	341	552	NA	NA	70.4	71.6	Oral	PCR-RFLP	9
Fronhoffs, 2001 [39]	Germany	Caucasian	HB	291	300	59.8	47.1	80.1	58	Oral, pharyngeal, laryngeal	RT-PCR	6
Hahn, 2002 [42]	Germany	Caucasian	PB	94	92	61.5	45.1	65.9	51.1	Oral	PCR-RFLP	7
Lei, 2002 [51]	China	Asian	HB	62	56	60.2	58.2	NA	NA	Laryngeal	PCR-RFLP	7
Varzim, 2002 [47]	Portugal	Caucasian	PB	88	172	62.8	43.0	94.3	72.7	Laryngeal	PCR-RFLP	7
Cheng, 2003 [49]	Taiwan	Asian	HB	279	325	NA	NA	NA	NA	Pharyngeal	PCR-RFLP	6
Gajecka, 2005 [40]	Poland	Caucasian	HB	289	311	57.9	45.9	100	100	Laryngeal	PCR-RFLP	8
Rydzanicz, 2005 [45]	Poland	Caucasian	HB	266	143	61.6	53.1	95.1	100	Oral, pharyngeal, laryngeal	PCR-RFLP	8
Unal, 2005 [46]	Turkey	Caucasian	HB	45	104	53.5	50.0	93.3	65.4	Laryngeal	PCR-RFLP	7
Marques, 2006 [58]	Brazil	Mixed	HB	231	212	56.6	55.3	83.5	79.2	Oral	PCR-RFLP	8
Gara, 2007 [57]	Tunisia	Mixed	HB	64	160	50.7	53.6	65.6	45	Oral, pharyngeal, laryngeal	PCR-RFLP	7
Majumder, 2007 [53]	India	Asian	HB	297	342	NA	NA	NA	NA	Oral	PCR-RFLP	6
Boccia, 2008 [22]	Italy	Caucasian	HB	210	245	63.6	63.3	71.4	72.2	Oral, pharyngeal, laryngeal	PCR-RFLP	8
Buch, 2008 [56]	USA	Mixed	PB	182	399	58.7	58.7	87.4	75.7	Oral	PCR-RFLP	9
Harth, 2008 [43]	Germany	Caucasian	HB	312	300	59.7	47.2	80.4	58.7	Oral, pharyngeal, laryngeal	PCR-RFLP	6
Chatzimichalis, 2010 [37]	Greece	Caucasian	HB	88	102	66.5	62.5	87.5	74.5	Laryngeal	PCR-RFLP	8
Demokan, 2010 [38]	Turkey	Caucasian	PB	95	93	59.6	53.3	86.3	52.7	Oral, pharyngeal, laryngeal	PCR	8
Hou, 2011 [50]	China	Asian	PB	172	170	49.6	49.6	100	100	Oral, pharyngeal	PCR-RFLP and Taqman	9
Balaji, 2012 [48]	India	Asian	HB	157	132	53.1	55.1	54.8	34.8	Oral	Taqman	7
Majumder, 2012 [52]	India	Asian	HB	299	381	NA	NA	NA	NA	Oral	PCR	6
Tian, 2013 [55]	China	Asian	PB	233	102	60.0	60.0	NA	NA	Laryngeal	PCR	8
Marques, 2014 [59]	Brazil	Mixed	PB	101	141	NA	NA	NA	NA	Oral, pharyngeal, laryngeal	PCR-RFLP	7

Abbreviations: HB, hospital-based; PB, Population-based; PCR, Polymerase Chain Reaction; RT, Real Time; RFLP, Restriction Fragment Length Polymorphism; NA, Not Available. Taqman: The 5′ Nuclease Assay.

**Table 2 medicina-57-01095-t002:** Prevalence of the polymorphisms of *N-acetyltransferases 1* and *2* (*NAT1* and *NAT2*), (slow vs. rapid acetylators).

Author, Year	* NAT1 *			
	Case		Control	
	Slow	Rapid	Slow	Rapid
Katoh, 1998 [35]	9	53	46	76
Henning, 1999 [23]	144	109	232	164
Jourenkova-Mironova, 1999 [44]	141	109	98	74
Olshan, 2000 [60]	83	88	108	85
Fronhoffs, 2001 [39]	195	96	206	94
Varzim, 2002 [47]	48	40	107	65
Demokan, 2010 [38]	53	42	42	51
Majumder, 2012 [52]	128	171	168	213
Author, Year	*NAT2*			
	Case		Control	
	Slow	Rapid	Slow	Rapid
Gonzalez, 1998 [41]	28	47	37	163
Katoh, 1998 [35]	7	55	7	115
Henning, 1999 [23]	138	117	286	224
Jourenkova-Mironova, 1999 [44]	142	108	91	81
Morita, 1999 [54]	18	127	17	147
Chen, 2001 [36]	198	143	302	250
Hahn, 2002 [42]	59	35	57	35
Lei, 2002 [51]	50	12	34	22
Varzim, 2002 [47]	47	41	76	96
Cheng, 2003 [49]	39	240	54	271
Gajecka, 2005 [40]	127	162	165	146
Rydzanicz, 2005 [45]	131	135	72	71
Unal, 2005 [46]	15	30	7	97
Marques, 2006 [58]	29	202	38	174
Gara, 2007 [57]	33	31	59	101
Majumder, 2007 [53]	190	107	205	137
Boccia, 2008 [22]	109	101	128	117
Buch, 2008 [56]	84	98	224	175
Harth, 2008 [43]	189	123	181	119
Chatzimichalis, 2010 [37]	39	49	65	37
Demokan, 2010 [38]	50	45	45	48
Hou, 2011 [50]	46	126	33	137
Balaji, 2012 [48]	100	57	67	65
Tian, 2013 [55]	189	44	56	46
Marques, 2014 [59]	48	53	51	90

**Table 3 medicina-57-01095-t003:** Subgroup analyses of association between *N-acetyltransferases 1 and 2* (*NAT1* and *NAT2*) polymorphisms and the risk of head and neck cancer (slow vs. rapid acetylators).

Polymorphism	Variable (N)	OR	95% CI	*p*-Value	I^2^	P_heterogeneity_
*NAT1*	Overall (8)	0.89	0.77, 1.02	0.09	48%	0.06
	Ethnicity					
	Caucasian (5)	0.96	0.80, 1.15	0.64	0%	0.45
	Asian (2)	0.55	0.17, 1.80	0.32	87%	0.005
	Control source					
	Hospital-based (6)	0.87	0.74, 1.01	0.06	46%	0.10
	Population-based (2)	1.05	0.51, 2.17	0.90	72%	0.06
	Sample size					
	≥200 (6)	0.90	0.77, 1.04	0.15	0%	0.87
	<200 (2)	0.67	0.13, 3.56	0.64	91%	0.0007
	Genotyping method					
	PCR (4)	0.94	0.79, 1.14	0.54	26%	0.26
	PCR-RFLP (3)	0.64	0.34, 1.18	0.15	74%	0.02
	Tumor type					
	Oral (2)	0.55	0.17, 1.80	0.32	87%	0.005
	Laryngeal (2)	0.87	0.67, 1.15	0.33	0%	0.43
*NAT2*	Overall (25)	1.22	1.02, 1.46	0.03	74%	<0.00001
	Ethnicity					
	Caucasian (13)	1.10	0.89, 1.37	0.38	71%	<0.0001
	Asian (8)	1.60	1.13, 2.26	0.008	69%	0.002
	Mixed (4)	1.04	0.61, 1.77	0.89	79%	0.003
	Control source					
	Hospital-based (15)	1.10	0.88, 1.37	0.39	71%	<0.0001
	Population-based (10)	1.41	1.04, 1.92	0.03	75%	<0.0001
	Sample size					
	≥200 (20)	1.19	1.00, 1.42	0.05	70%	<0.00001
	<200 (5)	1.49	0.68, 3.29	0.32	85%	<0.0001
	Genotyping method					
	PCR (4)	1.47	0.77, 2.78	0.24	85%	0.0002
	PCR-RFLP (19)	1.14	0.93, 1.39	0.21	72%	<0.00001
	Tumor type					
	Oral (7)	1.05	0.80,1.38	0.72	62%	0.01
	Pharyngeal (2)	0.82	0.54, 1.24	0.35	0%	0.96
	Laryngeal (8)	1.48	0.88, 2.51	0.14	88%	<0.00001

Abbreviations: PCR, Polymerase Chain Reaction; RFLP, Restriction Fragment Length Polymorphism.

**Table 4 medicina-57-01095-t004:** Meta-regression analysis of association between *N-acetyltransferases 1 and 2* (*NAT1* and *NAT2*) polymorphisms and the risk of head and neck cancer (slow vs. rapid acetylators).

Polymorphism	Variable		Point Estimate	Standard Error	Lower Limit	Upper Limit	Z-Value	*p*-Value
*NAT1*	Publication year	Slope	0.01830	0.01361	−0.00837	0.04497	1.34462	0.17875
		Intercept	−36.77098	27.26207	−90.20365	16.66169	−1.34880	0.17740
	Sample size	Slope	0.00027	0.00045	−0.00060	0.00115	0.61240	0.54027
		Intercept	−0.25993	0.24912	−0.74819	0.22833	−1.04340	0.29676
	Mean age of cases	Slope	−0.01179	0.03248	−0.07546	0.05186	−0.36300	0.71660
		Intercept	0.57037	1.93376	−3.21972	4.36047	0.29496	0.76803
	Mean age of controls	Slope	−0.02263	0.03624	−0.09365	0.04839	−0.62459	0.53224
		Intercept	1.17938	2.13386	−3.00290	5.36167	0.55270	0.58047
	Male percentage of cases	Slope	−0.01131	0.01256	−0.03593	0.01331	−0.90074	0.36773
		Intercept	0.86738	1.11137	−1.31087	3.04562	0.78046	0.43512
	Male percentage of controls	Slope	−0.00268	0.00617	−0.01478	0.00942	−0.43474	0.066375
		Intercept	0.03230	0.43459	−0.81948	0.88409	0.07433	0.94074
*NAT2*	Publication year	Slope	0.00944	0.01016	−0.01047	0.02934	0.092942	0.35267
		Intercept	−18.82284	20.36308	−58.73373	21.08806	−0.92436	0.35530
	Sample size	Slope	−0.00080	0.00020	−0.00120	−0.00040	−3.91239	0.00009
		Intercept	0.50882	0.11300	0.28733	0.73030	4.50265	0.00001
	Mean age of cases	Slope	−0.04050	0.01356	−0.06706	−0.01393	−2.098776	0.00281
		Intercept	2.47888	0.80007	0.91077	4.04699	3.09832	0.00195
	Mean age of controls	Slope	−0.00438	0.00889	−0.02180	0.01305	−0.49203	0.62270
		Intercept	0.34691	0.47403	−0.58217	1.27600	0.73184	0.46427
	Male percentage of cases	Slope	−0.0629	0.00393	−0.01399	0.00141	−1.60201	0.10915
		Intercept	0.57366	0.33428	−0.08152	1.22884	1.71610	0.08614
	Male percentage of controls	Slope	−0.00785	0.00289	−0.01351	−0.00219	−2.71989	0.00653
		Intercept	0.64373	0.22152	0.20956	1.07790	2.90598	0.00366

## Data Availability

No new data were created or analyzed in this study. Data sharing is not applicable to this article.

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
