# Peer review of "Association of *N-acetyltransferases 1 and 2* Polymorphisms with Susceptibility to Head and Neck Cancers—A Meta-Analysis, Meta-Regression, and Trial Sequential Analysis"

_medicina, 2021, doi:10.3390/medicina57101095_

Round 1

Reviewer 1 Report

The present study reports some interesting meta-analytic data of N-acetyltransferase-1/2 (NAT1/2) genes regarding polymorphic genetic diversity in slow and rapid acetylation phonotypes of head/neck cancers (HNCs). Any relationship between NAT1/2 polymorphism and HNC seems to be a clue to prediction . The 4 available D/Bs such as PubMed/Medline, Web of Science, Scopus, and Cochrane were searched for the purpose. The odds ratio (OR) with 95% confidence interval (CI) are analyzed by Review Manager program. The Comprehensive Meta-Analysis version 2.0 (CMA 2.0) software and trial sequential analysis (TSA) were used for analysis. 28 papers on NAT1 polymorphism and 25 on NAT2 polymorphism are used for the meta-analysis. Slow acetylation group of NAT2 polymorphism corresponds to high risk for HNC OR. Ethnicity, source, and genotyping as factors are associated with HNC risk-related NAT2 polymorphism.  From TSA prediction, more large scaled cases are suggested for meaningful associations. From the results, slow acetylators found in NAT2 polymorphism are associated with a HNC risk, but without relation  between NAT1 polymorphism and HNC. The findings are interesting in the specific readers of the field. 

The present study is suffered from any lacking of the positional variants due to multiple genetic polymorphism in the NAT1 and NAT2 genes. Thus, they have to show the genetic sequence specific for polymorphic region.

In addition, the NAT targeting substrates should be discussed with the polymorphisms and ethnic variations. In fact, the number of cases are too limited, requiring increased number of cases.

In conclusion the study is interesting in the field and recommend the extensive revision, incorporating the above basic question.

Author Response

We thank Reviewer #1 for their helpful and valuable comments. Please find the detailed point-by-point-response attached as a separate file. Again, thank you very much for all your kind efforts.

Reviewer 2 Report

Dear Authors,

The article: 'Association of N-acetyltransferases 1 and 2 polymorphisms with susceptibility to head and neck cancers – a meta-analysis, meta-regression, and trial sequential analysis' was  to update the evidence on the association between NAT1 and NAT2 polymorphisms and the risk of HNC by including more studies.

The article should be prepared according to the MDPI guidelines.

 There are punctuation mistakes to correct. 

Figure 1 should be in the Materials and Methods section.

To sum up, article can be accepted after corrections.

Author Response

We thank Reviewer #2 for their helpful and valuable comments. Please find the detailed point-by-point-response attached as a separate file. Again, thank you very much for all your kind efforts.

Round 2

Reviewer 1 Report

They authors have appropriately answered and revised the original version.

I am satisfied with the revision.

Reviewer 2 Report

Authors corrected the article.

To sum up, article can be accepted after Editors decision.

This manuscript is a resubmission of an earlier submission. The following is a list of the peer review reports and author responses from that submission.